# Fatigue Prediction of Aluminum Alloys Considering Critical Plane Orientation under Complex Stress States

**DOI:** 10.3390/ma13173877

**Published:** 2020-09-02

**Authors:** Marta Kurek

**Affiliations:** Department of Mechanics and Machine Design, Opole University of Technology, 5 Mikołajczyka Street, 45-271 Opole, Poland; ma.kurek@po.edu.pl

**Keywords:** fatigue life, aluminium alloys, critical plane

## Abstract

This publication is intended to present a new way of estimating the fatigue life of various construction materials. Carpinteri’s proposal was modified by replacing the fatigue limits ratio with the value of the normal to shear stress ratio for a given number of cycles. In this study, the proposed criterion and calculation model was verified for the selected group of aluminium alloys. The purpose of the analysis of the experimental studies was to check the effectiveness of the proposed method of estimating fatigue life under the applied bending and torsional load conditions. The results of the fatigue calculations are presented in graphical form by means of diagrams showing the comparison of design and experimental strength. Before fatigue life was calculated, the critical plane orientation according to Carpinteri’s model and the proposed model were determined. After analyzing the results of the comparison of design and experimental durability, it can be stated that the proposed fatigue life estimation algorithm gives satisfactory results for multiaxial cyclic loads.

## 1. Introduction

Currently, the approach to assessing the fatigue life of machinery and equipment components is similar in many areas of civil engineering. Examples include the design of airplanes and rockets, road and rail vehicles, ships and offshore drilling platforms, turbines, land structures, chemical and process equipment, working machinery and other equipment. The proper estimation of the fatigue life of these facilities is a very important problem of modern technology, and incorrect assessments can be the cause of disasters [1,2,3].

The phenomenon of fatigue has been a current topic of research on structural materials for nearly two hundred years, due to fatigue failure, which causes enormous material losses and an increase in the safety risk for working people every year. Many tragic accidents were caused by the appearance of fatigue cracks in structural elements, which in effect made it impossible to carry the loads and led to complete destruction of the entire structure. Therefore, the proper aim of all research conducted on the phenomenon of fatigue is to design and construct safe components of structures and machines.

The complex nature of fatigue processes has resulted in a large number of fatigue hypotheses, which are the basic tools in fatigue life prediction. These hypotheses reduce the dimensional stress state to an equivalent uniaxial stress state. The criteria of multiaxial fatigue originating in the above-mentioned hypotheses can be divided according to the physical nature of the failure parameter into stress criteria, strain criteria, and energy (stress-strain) criteria.

Stress formulation is applied for a large number of cycles, strain formulation is most often used for a small number of cycles, and energy formulation allows the components of stress and strain state to be taken into account simultaneously and can be used for both a small and large number of cycles.

The problem of estimating the fatigue life of various materials is not new. With the development of research on the phenomenon of fatigue, many models have been developed to estimate fatigue life [4,5,6]. A number of steps have been taken to define an algorithm to determine with the greatest possible accuracy the durability of machines or equipment subjected to alternating loads. The first work on the algorithm for estimating fatigue life under random loads in terms of stress was started by Prof. Macha in 1979 [7]. The algorithm developed by Macha for the general dimensional state of stress has already been modified many times and presented in various works [8,9,10,11,12].

Safety regulations and environmental protection standards force industries to look for the most efficient materials. Aluminium has been in ever-wider use from as early as the 1970s. Due to their characteristics, aluminium alloys are slowly replacing popular structural materials such as steel, plastic, and copper.

In the subject literature, we can also find a number of works dealing with various aspects of forecasting the fatigue strength of aluminium alloys [13,14,15]. Fatigue strength properties of pure aluminium are relatively low, hence the use of alloys that can be even several times as strong when adequately heat-treated. Aluminium alloys have a desirable structural parameter, i.e., the ratio of fatigue strength to specific weight, which is higher than that of steel. Furthermore, their impact strength does not decrease as temperature is lowered, so they exhibit a higher impact strength in lower temperatures than steel. Aluminium alloys are being increasingly used in the production of either machine components or entire structures working under operating loads that are not always of constant amplitude.

In [16], the author summarized the attempt at evaluating the fatigue strength of various materials subjected to proportionate and disproportionate loads. One of the materials was the LY12CZ aluminium alloy; in this case, the fatigue strength was calculated using a method based on the critical plane concept, with the stress-correlated factor taken into account. Article, where the authors study pipe couplings made of aluminium alloys A1Si1MgMn (EN AW 6082 T6) and AlMg3.5Mn (EN AW 5042) under disproportionate load conditions is worth noticing as well [17]. The fatigue behavior of elements was evaluated using the notch stress concept with the reference radius r_ref_ = 0.05 mm, while equivalent stresses were determined based on the stress space curve hypothesis. In [18], readers will find a broad analysis of crack surface changes and micro-mechanism formation in relation to both particle topology and the orientation and size of grains in the 7075-T651 rolled alloy. A new fatigue model for aluminium–silicon alloys with a low copper content was put forward in [19]. Renault has adopted a methodology for estimating the fatigue strength while taking the effect of thermal aging into account. It is also worth noting that aluminium alloys have found application in the selective laser melting (SLM) method. In this case as well, the fatigue strength is estimated [20,21] and crack propagation is observed [22]. A new proposition for estimating the fatigue strength, based on factoring in the variability in material parameters, was presented in [10,23]. In these works, the models were verified based on the findings from studies on fatigue in aluminium alloys.

The analysis of the resulting hypotheses shows that there is no universal algorithm to determine the fatigue life, taking into account both the type of material of the tested item and the nature of the load. These algorithms are often modified due to particular fatigue dependent variables. However, there are still difficulties in determining the fatigue life resulting from the impossibility to determine it using a single equation. The main advantage of the theoretical models is the significant time saving in trying to determine the fatigue life of materials or structures exposed to random loads without performing long-term experimental studies.

An additional advantage is that theoretical models can be used in reverse analysis in this case to determine critical load conditions. Examples of application of theoretical models can also be found in other works [24,25,26,27,28].

It should also be mentioned that theoretical models are often used to optimize machines or structures [29,30].

The main objective of this work is to find a method that will allow estimation of the fatigue life of elements subjected to multiaxial loads already at the design and construction stage of machinery and equipment parts. The proposed new fatigue life estimation model will take into account the different angles of critical plane orientation for chosen aluminium alloys under multiaxial load conditions. Based on the results of experimental research, a new method of computing the angle of the critical plane orientation will be developed, which will be implemented into the algorithm for estimating the fatigue life of materials under a complex load condition.

The aluminium alloy group analyzed consists of six materials: 6082-T6 [31], 2017A-T4 [32], D-30 [33], and three types of Al-Zn-Mg type alloys in grade 7003 for plastic working after the low-temperature heat-plastic treatment [34,35,36]. The results were used to analyze the correctness of the proposed model of fatigue life estimation.

Aluminium alloys are characterized by a very good specific strength, i.e., the ratio of tensile strength to density. Due to their properties, especially their lightness, they are used in structures in which the weight of the structure is an important factor, namely in aircraft, cars, rolling stock, energy industry, and construction, as well as the food and chemical industry. A drawback of aluminium alloys is their low melting point, which results in a rapid deterioration of mechanical properties with an increase in temperature. They are also characterized by very high plasticity, high electrical and thermal conductivity, good corrosion resistance, and high abrasion resistance.

## 2. Materials and Methods

### 2.1. Experimental Studies and Analysis of Literature Data

Alloys 6082-T6, 2017A-T4, and 7003 were tested in the laboratories of the Department of Mechanics and Machine Design on the fatigue machine MZGS-100 (Figure 1a) using “diabolo” type specimens (Figure 1b). The MZGS100 machine, which was designed and produced in Opole University of Technology (Opole, Poland) by Dr Achtelik, is used for the fatigue test of a material’s specimen subjected to cyclic loading as well as bending, torsion, and proportional combination of bending with torsion. It is possible to apply additional, static load representing the mean value of the load history. The MZGS-100 workstation consists of a drive system, a head, a loading system, and a control and measurement system. The system is driven by an AC electric motor with small power consumption (about 0.7 kW). The head with a clamp is fixed to the base and the rotating disk is mounted on the set of four flat springs fixed to the base as well. The specimen is fixed to the head using a clamp with screws. The other side of specimen is mounted in the clamp of the load lever. The load lever is connected with the rotating disk. The required load on the sample was obtained by balancing the rotating discs and setting the angle of rotation of the β lever, as graphically described in Figure 2. An additional, unbalanced weight on disk induces vibrations of the disk during rotation and its vertical displacement is transmitted to the load lever as force, which is the generated load moment of the specimen. Maximum value of the generated moment is M = 80 N·m and static load (mean value) can be set up to Mm = 60 N·m. Operation of the stand is based on inertial vibro-mechanism. The parameter controlled during testing was the progress of the total Mc moment applied to the specimen. The resultant moment M(t) can be decomposed to component moments: M_b_(t)—bending moment, M_t_(t)—torsion moment. At the beginning of testing, the test stand was upgraded with a new, improved control system of the bending moment on the sample by means of an external control system of the inverter settings used in the machine controller, coupled with a strain gauge bridge on the lever. The aim was to maintain a constant amplitude of the force moment, and thus the bending or torsional moment, acting on the tested specimen.

The value of the torsional moments *M_t_(t)* and bending moments *M_b_(t)* is related to the correlation as:(1)$$tg\beta =\frac{M_{t}\left(t\right)}{M_{b}\left(t\right)}$$

When *β = 0*, the specimen is bent, when *β = π/2*, the specimen is subject to torsion. In intermediate positions *0 < β < π/2*, both moments occur simultaneously according to the correlations:(2)$$M_{b}\left(t\right)=\left(M_{a}\sin\omega t\right)\cos\beta$$
(3)$$M_{t}\left(t\right)=\left(M_{a}\sin\omega t\right)\sin\beta$$

The result of both moments simultaneously is a state of stress in which the stresses *σ(t)* and *τ(t)* change their values in the phase and with the same frequency (proportional loads):(4)$$\sigma_{\beta }\left(t\right)=\sigma_{a\alpha }\sin\omega t$$
(5)$$\tau_{\beta }\left(t\right)=\tau_{a\alpha }\sin\omega t$$

The values of normal stresses *σ_β_(t)* and shear stresses *τ_β_(t)* within the elastic range can be determined:(6)$$\sigma_{\beta }\left(t\right)=\frac{M_{b}\left(t\right)}{W_{x}}$$
(7)$$\tau_{\beta }\left(t\right)=\frac{M_{t}\left(t\right)}{W_{0}}$$
where
(8)$$W_{x}=\frac{\pi d^{3}}{32}$$
(9)$$W_{0}=\frac{\pi d^{3}}{16}$$

Experimental tests were carried out on selected materials in the field of cyclic loads for pendulum bending, bilateral torsion, and the combination of bending and torsion. For the analysis of this study, the results of fatigue tests of duralumin D30 by Nishihara and Kawamoto [33] were also used.

In the case of the 7003 alloy, the material for the tests was an industrial aluminium alloy of Al-Zn-Mg type in the form of a sheet with dimensions 400 × 200 × 20 mm [34,35,36]. Al-Zn-Mg alloys of the 7000 series show the highest strength potential among alloys for precipitation hardening. Some of them contain Cu to improve their resistance to stress corrosion. The total content of Zn + Mg < 6% provides them with satisfactory resistance to cracking. The alloy was subjected to a low-temperature thermo-plastic treatment. Table 1 presents the chemical composition of the analyzed aluminium alloys while the basic mechanical parameters of the materials considered are presented in Table 2.

In the methodology assumed in the ASTM International norm [38], adopted for the analysis of experimental research results, the fatigue life is a value dependent on the amplitude of stress or strain. It is assumed that the distribution of fatigue life of samples obtained in the tests is a log-normal distribution of constant variance. The results of experimental studies for a large number of cycles were approximated by the regression equation for pendulum bending or tension–compression according to ASTM International recommendations:log *N_f_* = *A_σ_* + *m_σ_* log*σ_a_*(10)

For bilateral torsion or shear, the regression equation takes the form:log *N_f_* = *A_τ_* + *m_τ_* log*τ_a_*(11)
where *A_σ_, m_σ_, A_τ_, m_τ_* are the regression equation coefficients for pendulum bending or tension–compression bending and for bilateral torsion, respectively. The regression coefficients according to Equations (10) and (11) for particular load variants together with the determination coefficient *R^2^* are presented in Table 3.

### 2.2. General Model for Fatigue Life Estimation

For multiaxial loads, fatigue life computation is based on the reduction of the multiaxial load condition to its uniaxial equivalent state using appropriate fatigue strength criteria. The first criteria in history of multiaxial loading were: The criterion of maximum principal stress (Galileo), the criterion of maximum shear stress (Coulomb-Tresca-Guest), and the criterion of maximum octahedral stress (Huber-Mises-Hencky). These were proposals based on the principal stresses *σ*_1_, *σ*_2_, and *σ*_3_. Currently, in the subject literature there are many publications with new propositions of multiaxial fatigue criteria. In most cases, these criteria are dedicated to either groups of materials, types of loads, or manners in which loads are applied. It can be noted that propositions based on the critical plane concept constitute a large portion of the new criteria. The critical plane concept assumes that fatigue cracking of a material is the result of stresses in the (critical) plane of the material. The source of this assumption is the observation of cracks in metals that appear in certain planes. The concept of the critical plane concerns the initiation of crack, which is most often connected with the range of a high number of cycles (HCF).

This paper proposes a new model to estimate the fatigue life of various structural materials, taking into account the angle of orientation of the critical plane.

#### 2.2.1. Stress Computation

Stress state tensor component distribution:(12)$$\sigma_{xx}\left(t\right)=\sigma_{a}\sin\left(\omega t\right)$$
(13)$$\tau_{xy}\left(t\right)=\tau_{a}\sin\left(\omega t-\varphi \right)$$
where *σ_a_* is the amplitude of normal stress from bending, *τ_a_* is the amplitude of shear stress from torsion, *ω* is the angular frequency, *φ* is the phase shift angle, and *t* is time.

The input data in this model are stress values. In this study, normal stress amplitudes *σ_a_* from bending and shear stress amplitudes *τ_a_* from torsion were used for calculations. The stress values are calculated from Equations (14) and (15), respectively or taken from the literature and recalculated accordingly:(14)$$\sigma_{a}=\frac{M_{g}}{W_{x}}$$
(15)$$\tau_{a}=\frac{M_{s}}{W_{0}}$$
where *W_x_* is the cross-section bending modulus and *W*_0_ is the cross-section torsion modulus.

#### 2.2.2. Determination of the Angle of the Critical Plane Orientation and Calculation of the Equivalent Criterion

In the model presented herein, the method of damage cumulation was used to determine the angle of the critical plane orientation. In the model used, the failure is defined as the maximum value of the normal component, namely the normal stress. The progress of the normal stress oriented at an angle of *α* in respect to *σ_xx_* is given by the formula:(16)$$\sigma_{\eta }\left(t\right)=\;\sigma_{xx}\cos^{2}\alpha +\tau_{xy}\left(t\right)\sin2\alpha$$

The progress of shear stress, on the other hand, was formulated as:(17)$$\tau_{\eta s}\left(t\right)=-\frac{1}{2}\sigma_{xx}\left(t\right)\sin2\alpha +\tau_{x,y}\left(t\right)\cos2\alpha$$
(18)$$\alpha =\;\alpha_{\eta }+\beta$$
where *α_η_* is the maximum angle determined by the normal stresses, and *β* is the angle proposed by Carpinteri, which is determined in relation to the direction determined by the maximum in the normal direction as shown in Figure 3:(19)$$\beta =\frac{3}{2}\left[1-{\left(\frac{1}{B_{2}}\right)}^{2}\right]45{}^{\circ}$$
where *B*_2_ is the fatigue limit ratio expressed by the formula:(20)$$B_{2}=\frac{\sigma_{af}}{\tau_{af}}$$

In this paper, Equation (20) takes the form of the relation of normal stresses to shear stresses calculated for the assumed number of cycles, according to:(21)$$B_{2}=\frac{\sigma_{a}}{\tau_{a}}\left(N_{f}\right)$$

For further calculations, the criterion of maximum normal and shear stresses in the fracture plane was used in the form of [8]:(22)$$\sigma_{eq,a}=B\tau_{\eta s,a}+K\sigma_{\eta ,a}$$
where *σ_η_**_,a_ τ_η_**_s,a_* are the amplitudes of the normal and shear stresses in the chosen plane, respectively.

Most of the fatigue hypotheses presented in the literature address the random loads. However, each hypothesis prepared for multiaxial loads should also correctly describe the uniaxial states, i.e., bending or torsion (cyclic loads), on the basis of which parameters *B* and *K* were determined. The following considerations for bending have been made: *α**_η_* = 0, so *α* = *β*, and therefore Equations (16) and (17) take the form:(23)$$\sigma_{\eta ,a}=\;\sigma_{xx}\cos^{2}\beta$$
(24)$$\tau_{\eta s,a}=-\frac{1}{2}\sigma_{xx}\sin2\beta$$

Similarly, for torsion: *α**_ƞ_* = 45°, so *α* = 45° + *β*, so Equations (16) and (17) take the form:(25)$$\sigma_{\eta ,a}=\;\tau_{xy}\sin2\left(45{}^{\circ}+\beta \right)$$
(26)$$\tau_{\eta s,a}=\tau_{x,y}\cos2\left(45{}^{\circ}+\beta \right)$$

Inserting Equations (23)–(26) into (22) results in a system of equations, and after solving the equations for parameters *B* and *K:*(27)$$B=\frac{B_{2}-\frac{\sin\left(90{}^{\circ}+2\beta \right)}{\cos^{2}\beta }}{\frac{\sin2\beta \sin\left(90{}^{\circ}+2\beta \right)}{2\cos^{2}\beta }+\cos\left(90{}^{\circ}+2\beta \right)}$$
(28)$$K=\frac{2+B\sin2\beta }{2\cos^{2}\beta }=2-\frac{\sigma_{af}}{\tau_{af}}$$

The modified stress criterion in the plane of maximum normal and shear stresses, proposed as Equations (22), (27) and (28), was used for the calculations.

#### 2.2.3. Fatigue Life Calculation

In the case of constant-amplitude loads, fatigue life is determined by expressing the obtained amplitudes of the equivalent sequence as a function of the number of cycles. If the multiaxial state of stress is reduced to the state as for pure bending or tension–compression, the number of cycles is determined depending on the parameter [38] adopted for the description.

Fatigue life was calculated using Wöhler’s fatigue characteristics in accordance with the ASTM standard and formula:(29)$$logN_{f}=A-mlog\sigma_{aeq}$$
where *N_f_* is the number of cycles to failure, *A*, *m* are the regression equation coefficients for pendulum bending, and *σ_aeq_* is the amplitude of equivalent stress.

### 2.3. Fatigue Life Variation Analysis including Critical Plane Orientation Angle Change

The orientation of the critical plane should be understood as the orientation of the material point’s surroundings in space, not the plane of fatigue fracture. The direction of the critical plane can largely depend on the type of material. According to [39], materials in limit states can be elastic-brittle and elastic-plastic and exhibit intermediate properties, as in the case of aluminium alloys.

Manuscript [39] presents the dependence of changes of critical plane orientation on the ratio of fatigue limits for tension–compression and bilateral torsion according to different models based on Carpinteri’s and Spagnoli’s proposals. In earlier works by these authors [40,41], derivation of correlations (19) was not given; it was adopted arbitrarily. In addition, in the latest literature, there are attempts to determine the orientation of the critical plane, determining it through weight functions and using mean values of shear stresses [42]. However, in [43], the orientation of the critical plane is determined analytically with the assumption of a perfectly elastic body. Despite an increasing number of publications concerning the determination of the critical plane orientation angle, no consistent method for its determination has been found.

Therefore, on the basis of the available experimental studies, the appropriate model should be determined to correlate the determined angle with the ratio of normal stresses to shear stresses for both stress and strain models.

In this part of the analysis of fatigue life, the calculation of variability was performed in relation to the value of the *β* angle. Simulation studies were conducted in which it was assumed that *β*
*ϵ <0°,**4**5°>*, with discretization every 1°. For each of the 46 angles, the parameters *B* and *K* were calculated in accordance with (27) and (28). Figure 4 presents the variability of analyzed parameters depending on the value of *β* angle. Table 4 lists the value of parameter *K*, which is constant and independent of the *β* angle value.

Using a standard fatigue life estimation model, calculations were performed using a modified criterion in the plane of maximum normal stresses at each *β* angle for all the analyzed materials. These calculations were made only for the bending and torsion combination. Then, in order to check which *β* angle gives the closest results to those obtained for the experimental fatigue life, the fatigue life scatter analysis was carried out in accordance with the formula:(30)$$E=\sqrt{\frac{\sum_{i=1}^{n}log^{2}\frac{N_{exp}}{N_{cal}}}{n}}$$
where *n* is the number of samples taken for analysis, *N_exp_* is the experimental fatigue life, and *N_cal_* is the calculated fatigue life.

The final parameter used to evaluate the criterion is calculated from the following correlation:*T* = 10*^E^*(31)

The scatter was calculated for each angle *β* within the range *<0°,45°>* for each of the analyzed materials.

Figure 5 shows the value of *T* scatter depending on the value of the *β* angle for the analyzed materials. Each graph shows the minimum scatter and the angle at which this scatter was achieved.

One of the aims of this paper was to propose a new mathematical model for determining the angle of orientation of the critical plane. The initial assumptions concerned:-Considering the normal stress to shear stress ratio as a function of the number of cycles instead of the fatigue limit ratio.-Considering values less than 1 and greater than √3.-Using the *ctg* function, which perfectly fits into the analyzed correlation in the initial calculations.

On the basis of the presented analysis, the proposed own formulation for the *β* angle was proposed, which takes the form:(32)$$ctg\left(4\beta \right)=22.5{}^{\circ}\left[\frac{1+\sqrt{3}}{2}-\frac{\sigma_{a}}{\tau_{a}}\;\left(N_{f}\right)\right]for\;0{}^{\circ}\;\leq \beta \leq 45{}^{\circ}$$

### 2.4. Verification of the Proposed Model

An analysis of the proposed model is currently underway. It is planned to the perform analysis of the obtained *β* angle depending on the scatter. The second analysis will concern the change in the ratio of normal to shear stresses (previously marked as parameter *B*_2_).

#### 2.4.1. Analysis of the Obtained *β* angle Depending on the Scatter

The first of the conditions proposed by the author will concern the search for the optimal value of the *β* angle. For this purpose, the scatter values obtained from the calculations were used. It is assumed in these considerations that the scatter *T* is within the range from *T_min_* up to the scatter value increased by 10% of its value, marked *T*_1.1_, in accordance with
*T* ϵ <*T*_min_,*T*_1.1_>(33)

Figure 6 presents this assumption in graphic form.

The optimum value of the *β* angle obtained, according to the presented methodology, meets the following condition:(34)$$\beta_{opt}\;\in \;<\beta_{min,\;\;}\beta_{max}>$$

#### 2.4.2. Analysis of the Obtained *β* Angle Depending on the Fatigue Characteristics

The second analysis will concern the change in the ratio of normal to shear stresses (previously marked as parameter *B*_2_). Fatigue characteristics for simple states (bending or tension–compression and torsion) were used to calculate the fatigue life scatter (Figure 7).

According to the previous analysis, we know that:(35)$$B_{2}=\frac{\sigma_{a}\left(N_{f}\right)}{\tau_{a}\left(N_{f}\right)}$$

Subsequently, correlations for minimum and maximum values for the ratio of normal to shear stresses are proposed, using fatigue characteristics for simple states in accordance with the following:(36)$$B_{2max}=\frac{\sigma_{max}\left(N_{fi}\right)}{\tau_{min}\left(N_{fi}\right)}$$
(37)$$B_{2min}=\frac{\sigma_{min}\left(N_{fi}\right)}{\tau_{max}\left(N_{fi}\right)}$$

## 3. Results

Fatigue life calculations were performed to verify the compatibility of the determined fatigue life, determined using the new formula for the equivalent value to the fatigue life obtained by the experiment. The purpose of the analysis of experimental studies is to check the effectiveness of the proposed method of estimating fatigue life under the applied bending and torsional load conditions. The comparison was made for the bending/torsion combination. The analysis for bending and torsion was omitted because the values of the estimated life were the same for each of the analyzed angles. Before fatigue life was calculated, the critical plane orientation according to the Carpinteri model (according to 19) and the proposed model were determined. The graphs also show the results obtained using the β-angle from the scatter analysis carried out in the previous chapter. This is the value of the *β* angle obtained for the smallest scatter, denoted as *β_T_* and applies only to the bending and torsion combination.

Figure 8, Figure 9, Figure 10, Figure 11, Figure 12 and Figure 13 present a comparison of design and experimental fatigue life for the analyzed aluminium alloys. Each diagram also presents the values of *β* angles calculated from two correlations (19 and 32) and the value of the *β* angle obtained for the minimum scatter of *T* (Equation (31)) obtained from the analysis.

From the analysis of the graph presented in Figure 5, it can be observed that the fatigue life calculation results according to the Carpinteri model and the proposed model overlap. A similar situation can be observed in the case of results obtained using the *β* angle obtained from the analysis. The obtained results are very similar, most of them within the scatter band with the factor 3. It is justified by the very similar value of the gained value of the *β* angle, according to Carpinteri’s model *β =* 44°, and the author’s model *β* = 43°.

## 4. Discussion

Any new model or formula requires verification, during which the assumptions made should be confirmed. In order to verify the model, the scatter analysis of the estimated fatigue life was used, both in respect to the proposed angle and the value of the ratio of normal stresses to shear stresses. As well as the scatter analysis, this paper proposes two conditions that will be used for model verification and further analysis of the obtained results according to Section 2.

In the group of aluminium alloys, it can be observed that the two analyzed models gave very similar fatigue life results. For two aluminium alloys 2017A-T4 and 6082-T6, these computational points coincide with the results obtained for the *β_T_* angle. In other cases, the results calculated using Equations (19) and (32) are similar, but with a greater scatter than those obtained for *β_T_*.

Apart from the graphical verification of the proposed model, it is also necessary to calculate the fatigue life scatter for each of the analyzed computation methods. Using the formula for logarithmic correlations of experimental and computational fatigue life, the scatter was calculated for all analyzed materials and applied models, and the results are presented in graphic form in Figure 14.

In the case of alloys 7003-3.1 and 7003-4.2, only one value (*β* = 0°) meets this condition. This can be the cause of very high scattering and inconsistent results. In addition, we can link these incompatibilities to the plastic working. These alloys were additionally cold rolled in contrast to alloy 7003_P1.

For all aluminum alloys analyzed, calculations were carried out in accordance with the proposed conditions presented in Section 2.4. Using Equation (34), the optimum angle *β* was calculated and an example of the distribution for the selected material is shown in Figure 15. Table 5 summarizes detailed calculations concerning condition I.

By analyzing the presented graph and values from Table 5, it can be noticed that the scatter values vary depending on the value of the *β* angle. The assumption is that the scatter value can vary by up to 10%, taking a different form depending on the material. For example, for the 2017-T4 aluminium alloy, the assumption is fulfilled for angles *β* within the range β ϵ <4°,45°>. This means that the fatigue life results calculated for angles within this range will have similar values. With 7003-3.1 and 7003-4.2, only one angle value *β* = 0° satisfies this requirement.

The analysis for the proposed Condition II was conducted for all aluminium alloys and a detailed summary was presented in Table 6. Values of scatter for oscillatory bending (*T_σ_*) and bilateral torsion *(T_τ_)* were also included in the table.

Using the data presented in Table 5 and Table 6, in terms of the relationship between the *β* angle value and the ratio of normal to tangent stresses (*B*_2_), the fields were marked with where the obtained results of design fatigue life will be similar. The analysis for two selected materials is presented in Figure 16.

## 5. Conclusions

On the basis of a literature review, the analysis of experimental and literature research results and calculations, the following conclusions were made:Modification of the Carpinteri’s proposal by introducing the ratio of normal to shear stresses instead of fatigue limits in the case of the analyzed group of aluminum alloys has no significant effect on the results of the fatigue life. The average scatter for Carpinteri’s and the new proposed model is almost identical. For all analyzed materials the difference is less than 0.5%.In the case of alloys 7003_3.1 and 7003_4.2 (materials after plastic working), none of the models achieve satisfactory results, and the values of fatigue life scatter are between 6 and 7, which is too high for fatigue life prediction.The scatter of results for the remaining four aluminum alloys is in the range of 2 to 3. This value is considered a satisfactory result in the case of fatigue life estimation.The new β angle concept proposed by the author of this manuscript is based on the ratio of normal stresses to shear stresses and covers a wider range than *<1*;*√3>.* The value of this ratio often exceed this range for materials with non-parallel characteristics.The extended scatter analysis resulted in obtaining of safe ranges of the beta angle and the B_2_ parameter guaranteeing a similar fatigue life of the analyzed material.The presented mathematical model is dedicated to the cyclic loads under combination of bending and torsion and is intended to facilitate the work of engineers in estimating the fatigue life of machine elements under such loads.

## Figures and Tables

**Figure 1 materials-13-03877-f001:**
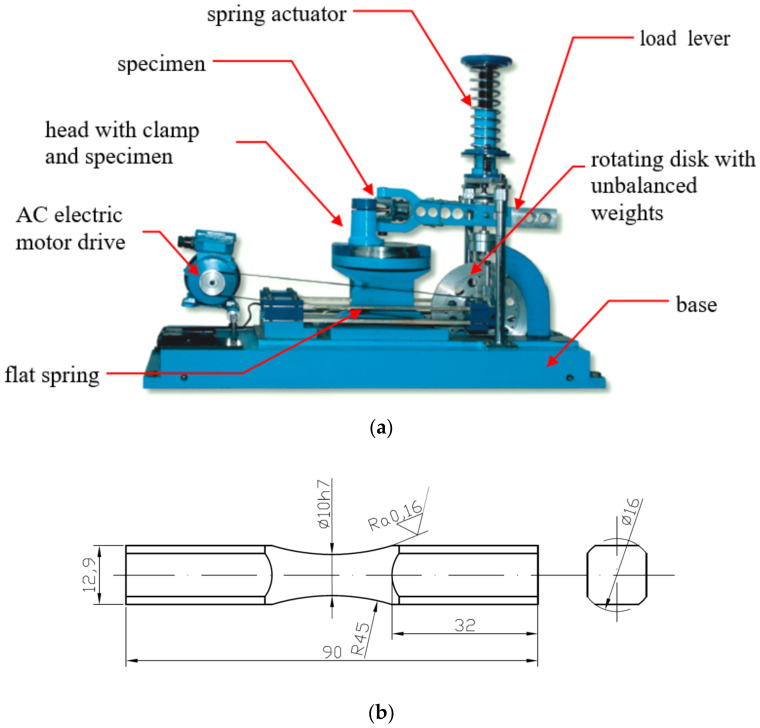
(**a**) Fatigue machine MZGS-100 [37]; (**b**) geometry of the samples used in the tests.

**Figure 2 materials-13-03877-f002:**
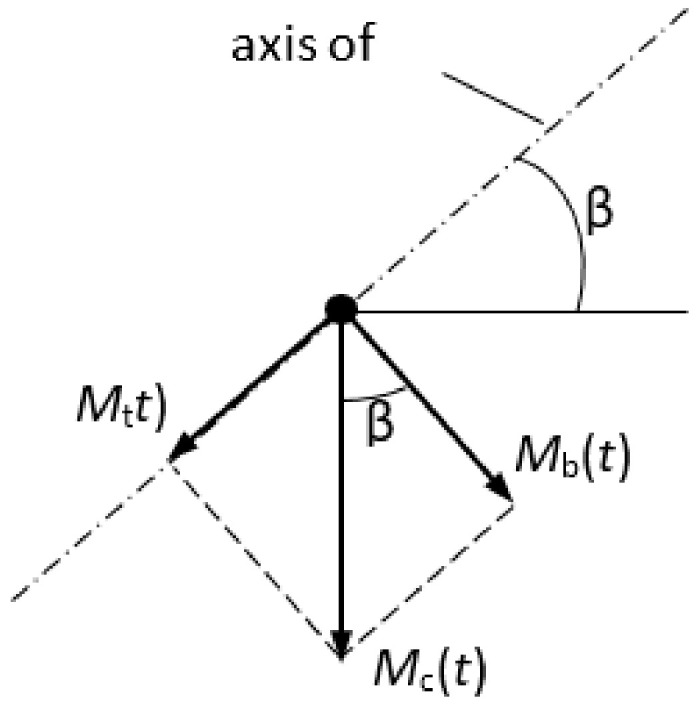
Interpretation of the lever angle *β*.

**Figure 3 materials-13-03877-f003:**
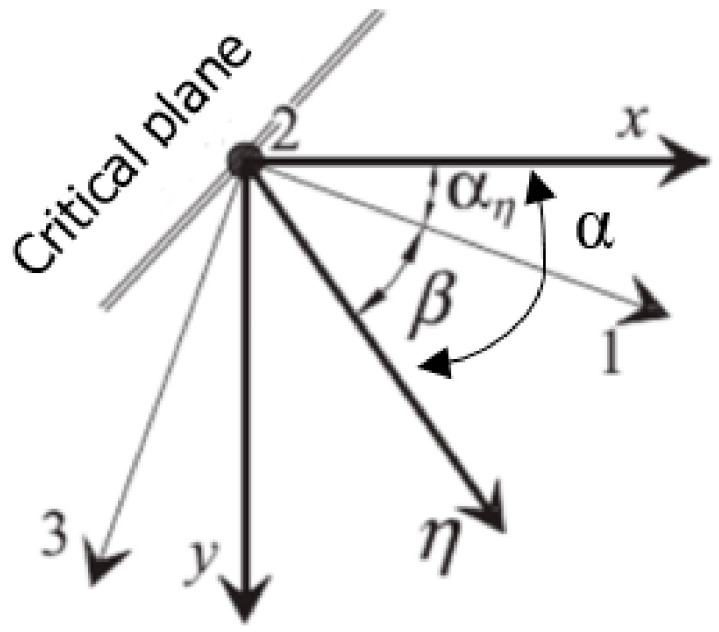
Graphical interpretation of the angle *β*.

**Figure 4 materials-13-03877-f004:**
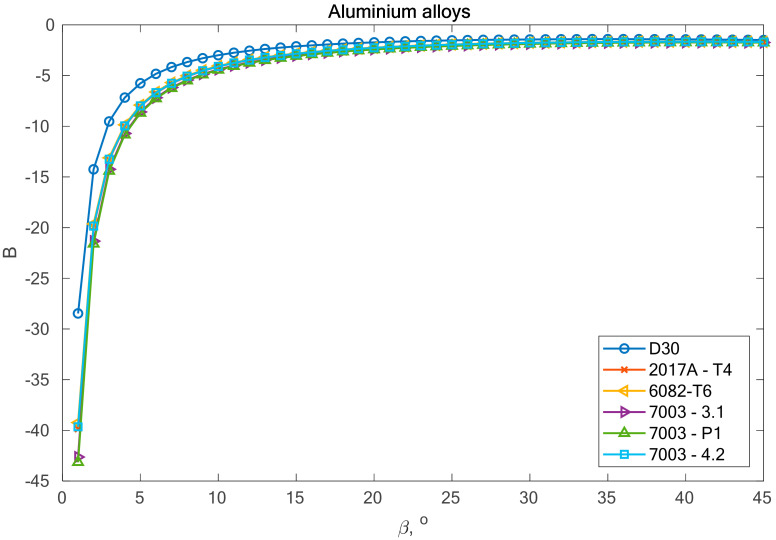
Variability of parameter *B* in relation to the value of *β* angle for analyzed aluminium alloys.

**Figure 5 materials-13-03877-f005:**
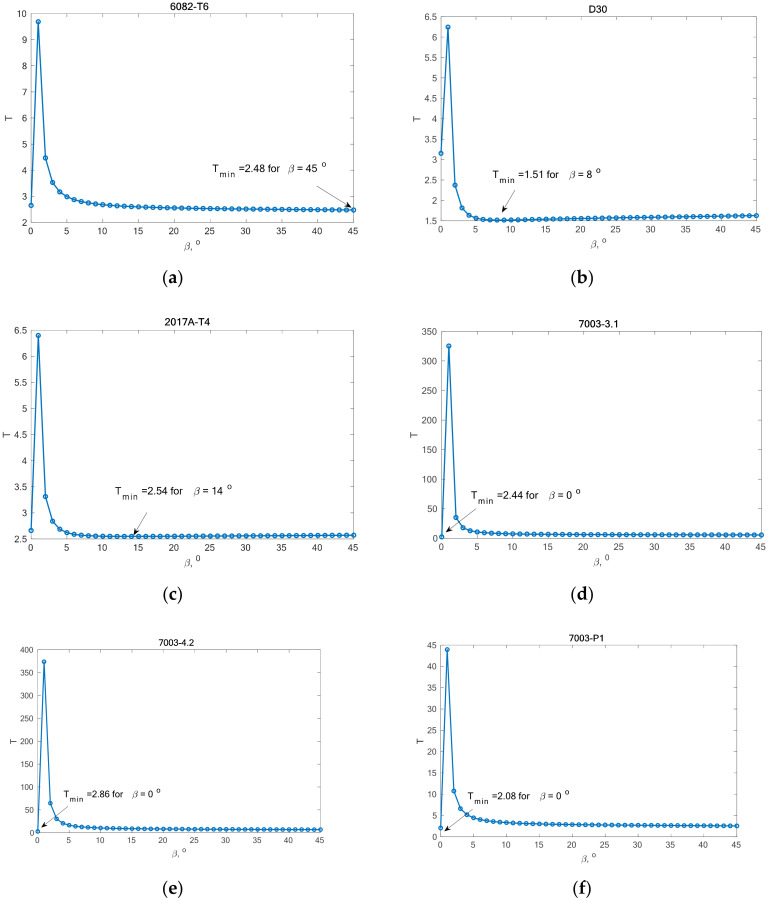
The correlation between T scatter and the β angle for the analyzed aluminium alloys: (**a**) 6082-T6; (**b**) D30; (**c**) 2017A-T4; (**d**) 7003-3.1; (**e**) 7003-4.2; (**f**) 7003-P1.

**Figure 6 materials-13-03877-f006:**
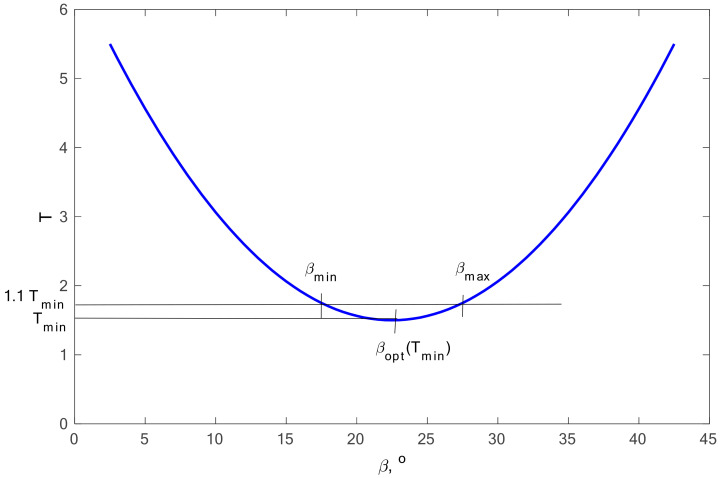
Methodology of presented assumptions—condition I.

**Figure 7 materials-13-03877-f007:**
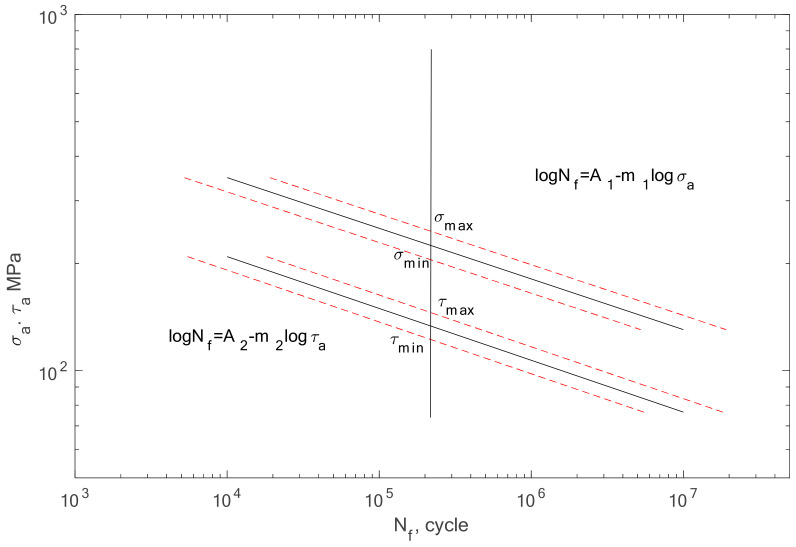
Methodology of presented assumptions—condition II.

**Figure 8 materials-13-03877-f008:**
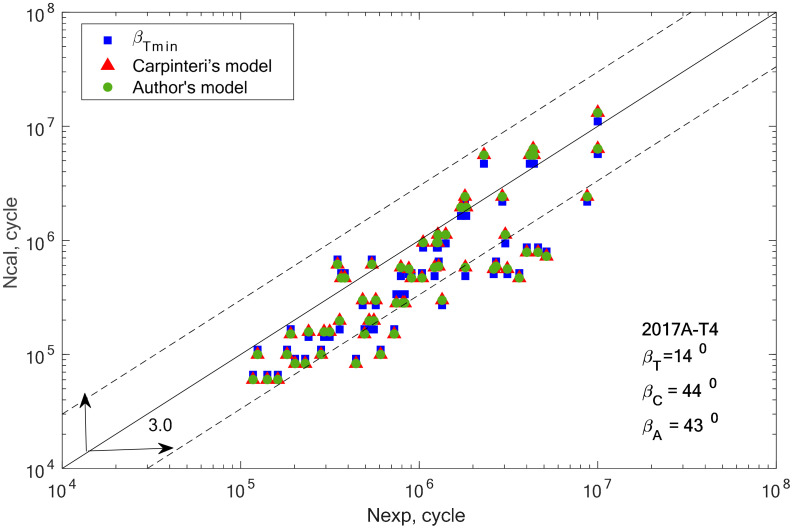
Comparison of the obtained design and experimental fatigue life for aluminium alloy 2017A-T4 for the cyclic bending–torsion combination.

**Figure 9 materials-13-03877-f009:**
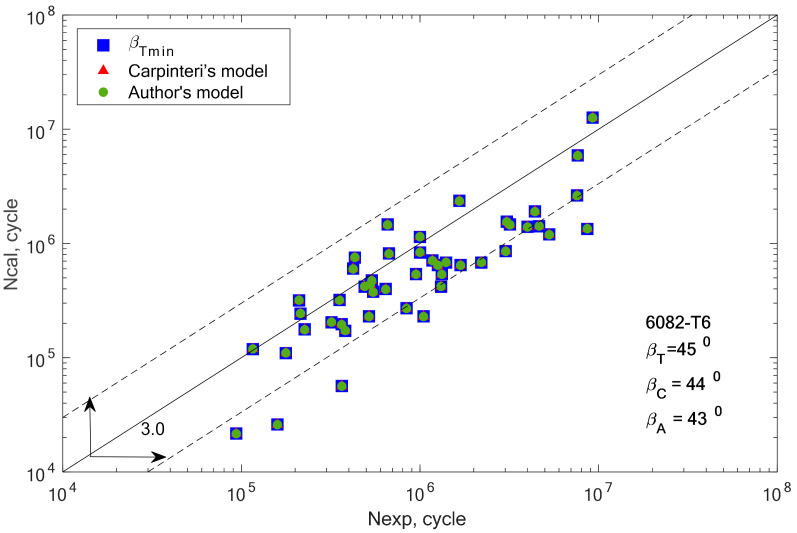
Comparison of the obtained design and experimental fatigue life for aluminium alloy 6082-T6 for the cyclic bending–torsion combination.

**Figure 10 materials-13-03877-f010:**
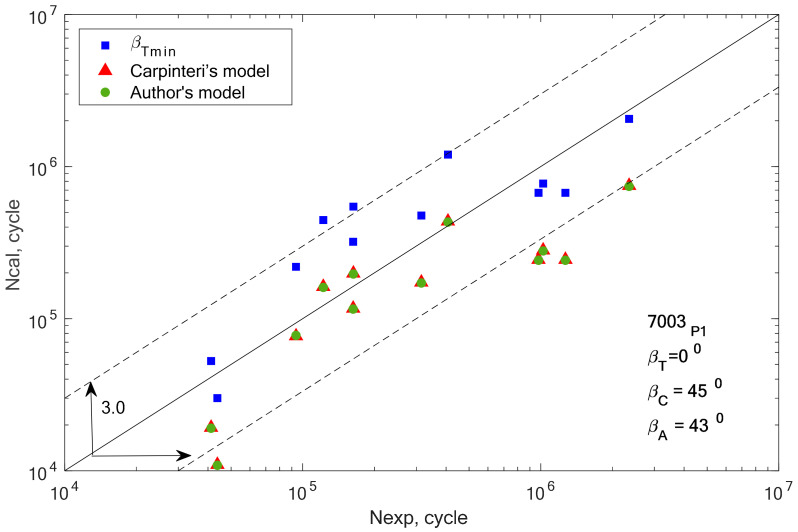
Comparison of the obtained design and experimental fatigue life for aluminium alloy 7003-P1 for the cyclic bending–torsion combination.

**Figure 11 materials-13-03877-f011:**
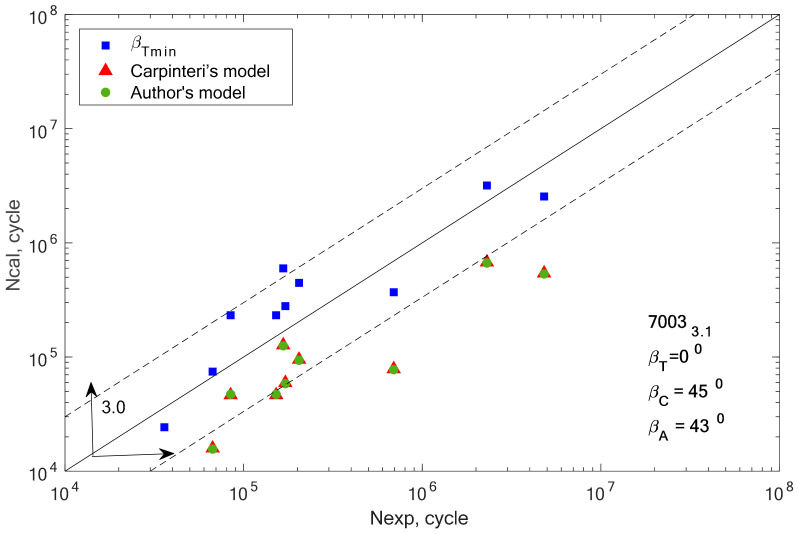
Comparison of the obtained design and experimental life for aluminium alloy 7003-3.1 for the cyclic bending–torsion combination.

**Figure 12 materials-13-03877-f012:**
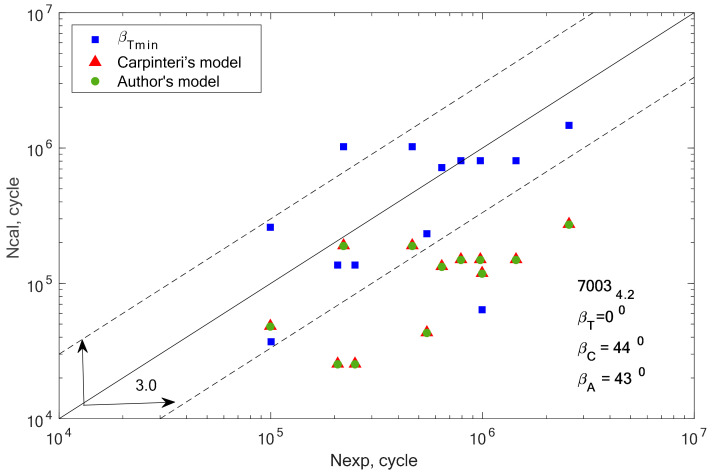
Comparison of the obtained design and experimental life for aluminium alloy 7003-4.2 for the cyclic bending–torsion combination.

**Figure 13 materials-13-03877-f013:**
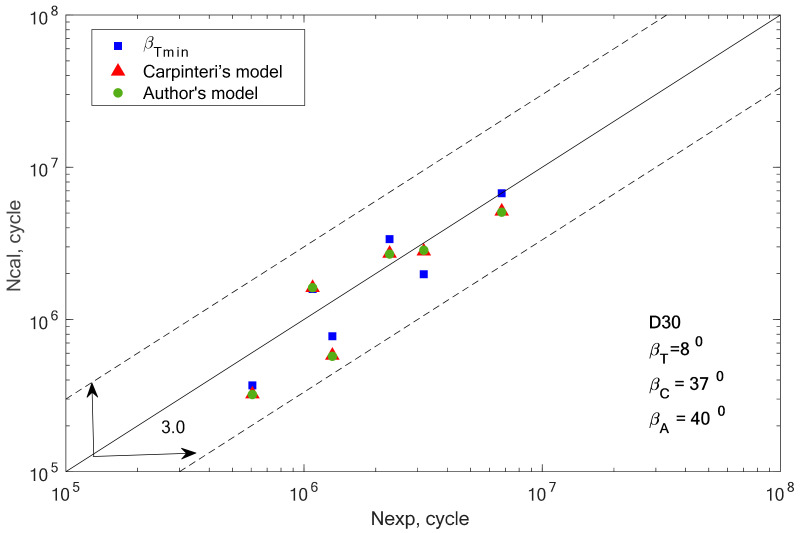
Comparison of the obtained design and experimental life for aluminium alloy D30 for the cyclic bending–torsion combination.

**Figure 14 materials-13-03877-f014:**
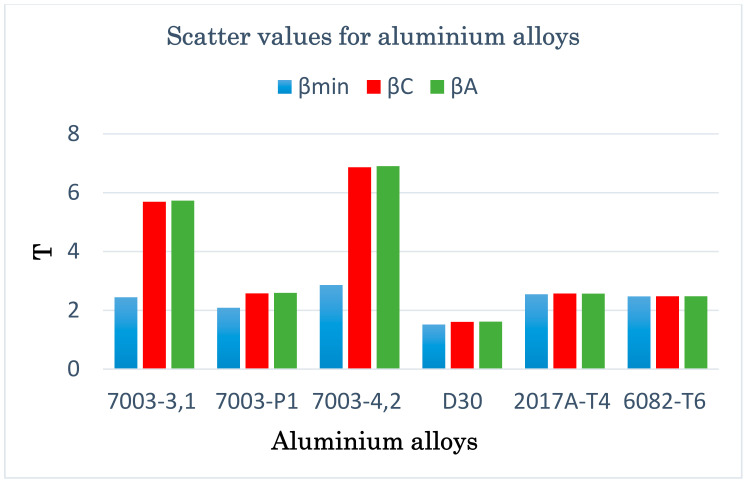
Scatter values for the design models used in the case of aluminium alloys.

**Figure 15 materials-13-03877-f015:**
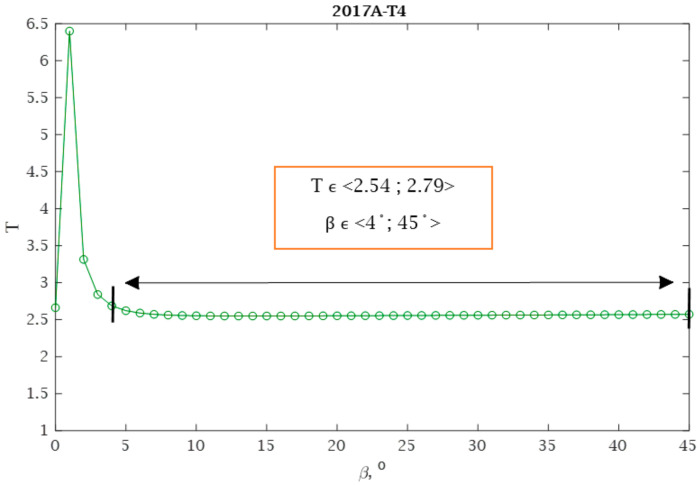
Correlation between *T* scatter value and *β* angle for aluminium alloy 2017A-T4.

**Figure 16 materials-13-03877-f016:**
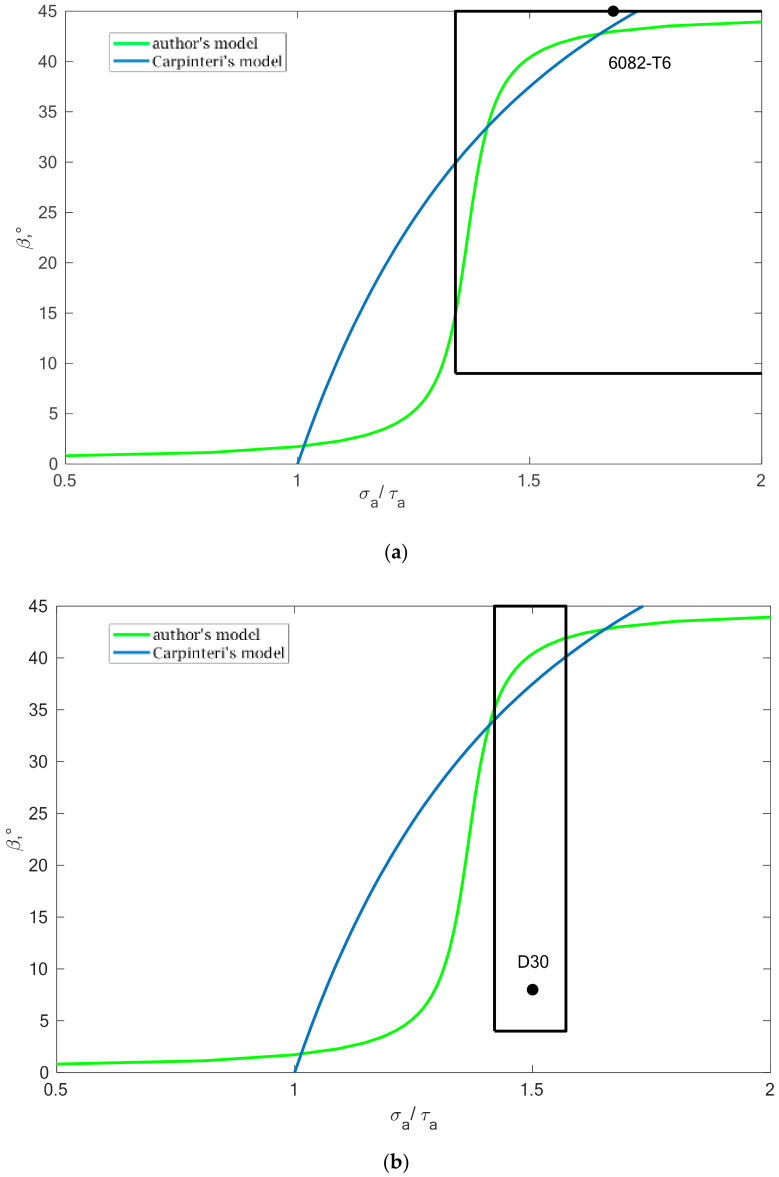
Secure area determined for (**a**) 6082-T6 and (**b**) D30 aluminium alloys in terms of the *β* angle in relation to the ratio of normal stresses to shear stresses.

**Table 1 materials-13-03877-t001:** Chemical composition of the analyzed materials in % (remainder Al).

Material	Mg	Cu	Zr + T	Zn	Mn	Fe	Si	Cr
6082-T6 [31]	0.6 ÷ 1.2	<0.1	<0.1	<0.2	0.4 ÷ 1.0	<0.5	0.7 ÷ 1.3	<0.25
2017A-T4 [32]	0.4–1.0	3.8–4.8	-	0.5	0.4–1.0	0.7	0.2–0.8	-
D-30 [33]	0.42	3.81	-	-	0.44	0.38	0.35	-
7003_P1 [34,35,36]	0.74	0.04	0.08	6.13	0.29	0.20	0.12	0.17
7003_3.1 [34,35,36]	0.74	0.04	0.08	6.13	0.29	0.20	0.12	0.17
7003_4.2 [34,35,36]	0.74	0.04	0.08	6.13	0.29	0.20	0.12	0.17

**Table 2 materials-13-03877-t002:** Basic mechanical parameters of the analyzed materials.

Material	R_e_, MPa	R_m_, MPa	A_12.5_%	ν	E, GPa
6082-T6 [31]	365	385	27.2	0.32	77
2017A-T4 [32]	395	545	21	0.32	72
D-30 [33]	306	437	14.3 *	0.32	72
7003_4.2 [34,35,36]	347	400	14.1 *	0.32	69
7003_P1 [34,35,36]	98.3	213.6	25.4 *	-	-
7003_3.1 [34,35,36]	256	321	10.2 *	-	-

* A_10._

**Table 3 materials-13-03877-t003:** Coefficients of regression together with the *R^2^* parameter of analyzed aluminium alloys.

Material	Bending			Torsion		
	*A_σ_*	*m_σ_*	*R^2^*	*A_τ_*	*m_τ_*	*R^2^*
2017A-T4 (PA6) [32]	21.8	−7.03	0.824	19.94	−6.87	0.876
D-30 [33]	30.54	−10.75	0.878	25.38	−9.17	0.975
6082-T6 [31]	23.8	−8.0	0.653	21.4	−7.7	0.696
7003_P1 [34,35,36]	24.6	−8.6	0.488	23.5	−9.1	0.432
7003_3.1 [34,35,36]	34.8	−13.14	0.668	27.7	−11.15	0.719
7003_4.2 [34,35,36]	39.5	−15.3	0.801	26.3	10.4	0.617

**Table 4 materials-13-03877-t004:** Summary of *K* values for aluminium alloys.

Group of Materials	Material	K
Aluminium alloys	6082-T6	0.315
	2017A-T4	0.304
	D30	0.503
	7003-P1	0.247
	7003-3.1	0.256
	7003-4.2	0.308

**Table 5 materials-13-03877-t005:** Details of condition I.

Group of Materials	Material	*T_min_*	*β_opt_*	*T_1,1_*	*β_(1,1)_*
Aluminium alloys	6082-T6	2.47	45°	2.72	9°–45°
	2017A-T4	2.55	14°	2.8	4°–45°
	D30	1.52	8°	1.67	4°–45°
	7003-P1	2.09	0°	2.29	0°, 23°–45°
	7003-3.1	2.44	0°	2.68	0°
	7003-4.2	2.86	0°	3.14	0°

**Table 6 materials-13-03877-t006:** Values of scatter for oscillatory bending *(T_σ_)* and bilateral torsion (*T_τ_*).

Group of Materials	Material	*T_σ_*	*T_τ_*	*B_2max_*	*B_2min_*	*B_2opt_*
Aluminium alloys	6082-T6	1.86	3.06	2.1	1.34	1.68
	2017A-T4	1.91	1.82	1.98	1.38	1.66
	D30	1.44	1.17	1.57	1.42	1.5
	7003-P1	2.02	2.4	2.04	1.48	1.79
	7003-3.1	1.54	2.37	1.95	1.56	1.74
	7003-4.2	1.77	1.69	1.85	1.55	1.69

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
