# Peer review of "Fatigue Prediction of Aluminum Alloys Considering Critical Plane Orientation under Complex Stress States"

_materials, 2020, doi:10.3390/ma13173877_

Round 1
Reviewer 1 Report
The authors ignored the comments.
The paper is still based on the Carpinteri criterion that is a very weak starting point.
The paper cannot be considered for pubblication.
Author Response
I respect your opinion.
In my opinion and many other scientists, this criterion can be successfully used in the assessment of fatigue life.
The article was corrected in accordance with the recommendations of other Reviewers.
Reviewer 2 Report
Dear author, thanks for submitting your paper dealing with the estimation of the fatigue life of aluminium alloys.
It is a quiet good paper and only a few issues have to be improved:
- Please use "." as decimal separator in equations, tables, diagrams and illustrations (partly it is mixed)
- Please replace Fig. 1a. The test machine is difficult to recognize and details should be marked in the figure to give the reader more information.
- The description of the test setup has to be improved. From the written text it is not fully clear. It is also written that the author is a co-executer of the tests, performed. Is it clear that the author has the exclusive rights to the publication of the results
- Please change Fig. 3 cause of poor quality
- The results are very good and show very good agreement between prediction and experiment for all alloys except 2. Please discuss more about the origin of these deviations in the discussion.
- Diagrams should be more or less in the same style with comparable font sizes, partly the axis captions are not readable.
- Lit 16: there is something wrong with the authors of the article. Please check
Author Response
I would like to thank you for careful review of the article and highlighting the weaknesses of the work. The article was corrected in accordance with the recommendations of all Reviewers. Some of the comments were overlapping with the other reviews. I tried my best to fulfil all comments as good as possible.
Please see the attachment

Reviewer 3 Report
This work presented a theoretical model for the prediction of fatigue life of aluminum alloys considering different materials and different loading conditions. The critical plane orientations were determined based on the normal to shear stress ratio in the modified Carpinteri’s model. The model calculations were validated with experimental results. Overall, the topic is suitable for the journal and worthy of investigation. The manuscript was well prepared with some interesting results presented. It can be accepted for publication after addressing the following issues.
- The title should be more concise. For example, “Fatigue Prediction of Aluminum Alloys considering critical plane orientation under complex stress states”.
- The contribution and significance of the presented model should be further discussed, in terms of the advantages of the theoretical model and usefulness in real applications. For example, the theoretical model can be employed with inverse analysis to determine critical loading conditions. Please review the following references for more information on inverse analysis. https://doi.org/10.1007/s00170-018-2508-6; https://doi.org/10.1007/s00170-019-03286-0; https://doi.org/10.1016/j.camwa.2007.09.003; etc.
- How did the authors to determine the normal stress to shear stress ratio? Experiments or Calculation?
- Where did the author validate the prediction accuracy of the presented model under different loading conditions (stress states)? It seems that the authors have validated the model calculations for different materials, and the original Carpinteri’s model has the comparable predictive capability.
- What are the limitations of the current work and corresponding future works for improvements? Please discuss more details. “5. Some materials have few experimental points, so further analysis of the proposed model is required for other construction materials.” is not rigorous and not acceptable.
- The conclusion is a summary of the section of results and discussion. The key findings should be highlighted with sufficient explanations to help readers better understand the contribution and significance of the current work.
- Please provide high-resolution images for figure 3.
- A nomenclature section is needed to properly define all symbols and abbreviations presented in the manuscript.
Author Response

(The authors gave the same response as above.)

Reviewer 4 Report
The article is devoted to estimating fatigue life for aluminum alloys.
The abstract of the paper is meaningful. I would suggest shortening the title if the author found it suitable.
The introduction is logic and presents a completed overview of the research domain.
The scientific novelty of the work is highlighted as the new fatigue life estimation model that has a few particular features related to the different angles of critical plane orientation for six aluminum alloys.
Lines 147-153 are not correctly placed. It is better to place them before the problem statement as it is more related to the general information, or at the beginning of section 2.
"the author was a co-executor of the tests." - I would suggest mentioning among the authors all co-authors that took part in this research. This phrase is not suitable for the article.
Figure 1a, if the picture of the machine can be found on the internet, there is no necessity to publish it. Moreover, this photo does not bring new data related to the research. I would suggest removing it or replacing it with a principal scheme of the machine.
All the equipment should be mentioned with a producer, city, and country of origin.
All the formulas should be mentioned with the source of origin.
The geometry of samples can be combined in one figure.
The chemical composition and mechanical properties should be mentioned with the source of origin if these data were not obtained during current research.
Lines 241-242, "In the methodology assumed in this book, adopted for the analysis of experimental research results" - this paper is a scientific article to be published or a book?
"ASTM," I would suggest reducing the number of acronyms in the paper or avoid them. Moreover, I did find these formulas there.
Lines 287-288, it repeats the previously mentioned sentence in section 1.
Lines 279-280, Line 297, it should be supported by references.
Figure 4, page 9. Is it a result figure of the research, or was it analytically calculated based on literature data?
Eq. 32, "???" - what is it?
I would suggest placing everything related to known data on methods and materials in Section 2 when everything that was proposed in Section 3 with the results. Section 3 should have as well subsections and be more completed.
Lines 609-610, it is not a discussion. You should not conduct research to say this phrase.
Lines 610-614, it hs no relation to the discussion of the results and repeats section 2.
Line 617-620, description of the results, there is no discussion.
Lines 621-625, there is a description of methods, after a short note on the results, but no discussion.
Lines 629-623, there is no discussion, plans for section 2.
Lines 662, from this line, the discussion starts.
Lines 669-678, there is no discussion, but plans for section 2.
Lines 678-686, there is the discussion.
Line 696, book?
"This is important for the analysis of materials with non-parallel characteristics" - it is a weak conclusion. It can not be just important. I would suggest making conclusions 1) more countable, 2) less oriented on the personality of the author, 3) more meaningful and fundamental. The first conclusions should be related to the developed model, and the second to the practical significance of the work. I would suggest finishing this article with some proposals on how the results of this study can influence the state of the Industry.
Fig. 12 need to be revised. 1) All the axis should have titles. 2) I would suggest changing the type of diagram if the author found this suggestion suitable.
Author Response

(The authors gave the same response as above.)

Reviewer 5 Report
I think the revised version is worth of publishing.
Author Response
Thank you for your comment.
Round 2
Reviewer 1 Report
My opinion is still stile negative.
Author Response
Thank you for your reply
Reviewer 2 Report
Dear author, thank you for improving the paper. From my point of view it can be published in the present form.
Regards
Author Response
Dear Reviewer
Thank you for your opinion.
Best regards
Reviewer 3 Report
The authors have addressed the raised issues. The reviewer has no further concern regarding the current work. It can now be accepted for publication.
Author Response

(The authors gave the same response as above.)

Reviewer 4 Report
The article was revised but a few important comments were ignored.
The most important that the article does not present a completed study.
1) In the conclusions we met again mentioning "book" - "4. The new β angle concept proposed by the author of this book is based on the ratio of normal
607 stresses to shear stresses and covers a wider range than <1; √3>". The author should decide is it a book or an article.
2) I would avoid saying the author's model, but the proposed model.
3) It has stayed undiscussed why the author affiliated in the Opole University of Technology referred to American standards, but not International, European, or national, that seems to be more logical.
4) At the same time, references stayed very national or from the same research group.
5) The comments from 18 to 26 were ignored.
6) The graphs stayed as it is with no titles and measuring units.
7) The text is written with plenty of miss-spellings, even in the grades of used materials.
8) The conclusions stayed uncountable with no emphasized novelty and practical significance for the industry.
Author Response
Dear Reviewer
Thank you for your opinion.
Please see the attachment.
Best regards

Round 3
Reviewer 4 Report
The article can be accepted fr publication in the present form. However, I would like to note that there is no "very good specific strength" (Line 130) in scientific work. I would recommend that it can be "good" in certain conditions for some particular applications (should be specified by the author) or at the required level of a countable parameter (also should be specified).